# Mitigating Health Data Poverty: Generative Approaches versus Resampling for Time-series Clinical Data

**Raffaele Marchesi** [†,1,2]    **Nicolo Micheletti** [†,1,3]    **Giuseppe Jurman** [1]    **Venet Osmani** [1]

```
raffaele.marchesi@studenti.unitn.it
nicolo.micheletti@student.manchester.ac.uk
jurman@fbk.eu, vosmani@fbk.eu
```

[†] Equal contribution

[1] Fondazione Bruno Kessler Research Institute, Trento, Italy
[2] University of Trento, [3] University of Manchester

## Abstract

Several approaches have been developed to mitigate algorithmic bias stemming from health data poverty, where minority groups are underrepresented in training datasets. Augmenting the minority class using resampling (such as SMOTE) is a widely used approach due to the simplicity of the algorithms. However, these algorithms decrease data variability and may introduce correlations between samples, giving rise to the use of generative approaches based on GAN. Generation of high-dimensional, time-series, authentic data that provides a wide distribution coverage of the real data, remains a challenging task for both resampling and GAN-based approaches. In this work we propose CA-GAN architecture that addresses some of the shortcomings of the current approaches, where we provide a detailed comparison with both SMOTE and WGAN-GP*, using a high-dimensional, time-series, real dataset of 3343 hypotensive Caucasian and Black patients. We show that our approach is better at both generating authentic data of the minority class and remaining within the original distribution of the real data.

## 1   Introduction

As machine learning methods increasingly weave themselves into societal decision making, critical issues related to decision fairness and algorithmic bias are coming to light. These issues are especially prominent in health and clinical decision making, where underprivileged and minority groups are underrepresented, resulting in unfair decisions. Algorithmic bias can originate from diverse sources, including health data poverty [1], where particular groups might be underrepresented in the training sets, but it may also originate from procedural care practices, wider socioeconomic issues or the data itself [2]. There are several attempts to address bias and improve fairness stemming from health data poverty. One approach is data augmentation, where synthetic data are generated from unbalanced datasets, mitigating minority class representation.

The Machine learning community has developed various approaches to generate synthetic data [3]. One of the widely used methods is data resampling, where the data from the minority class are typically oversampled to generate additional synthetic data, with Synthetic Minority Over-sampling TEchnique (SMOTE) [4] being a representative example. Synthetic samples lie between a randomly selected sample and its randomly selected neighbour (using k-NN), resulting in plausible samples close in feature space to the existing samples. SMOTE and related approaches are widely used due to their simplicity and computational efficiency. However, in high-dimensional data SMOTE may

NeurIPS 2022 Workshop on Synthetic Data for Empowering ML Research.

decrease data variability and introduce correlation between samples [5, 6, 7]. As such, alternative approaches based on generative adversarial networks (GAN) are gaining ground [8, 9, 10, 11, 12]. However, generation of high-dimensional time-series data remains a challenging task [13, 14, 15]. In this work we propose a new generative architecture, Conditional Augmentation GAN (CA-GAN), based on the Wasserstein GAN with Gradient Penalty [16, 17] as presented in Health Gym [18] (referred in this paper as WGAN-GP*), however with a different objective. Instead of generating new synthetic datasets, we focus on data augmentation, specifically augmenting the minority class to mitigate data poverty. We compare the performance of our CA-GAN with WGAN-GP* and SMOTE in augmenting data of patients of an underrepresented ethnicity (Black patients in our case), using a critical care dataset of 3343 hypotensive patients, derived from MIMIC-III database [19, 20].

**Contributions.** (1) We propose a new architecture CA-GAN for data augmentation, to address some of the shortcomings of the traditional and recent approaches in high-dimensional, time-series synthetic data generation. (2) We compare qualitatively and quantitatively CA-GAN with state of the art architecture in the synthesis of multivariate clinical time series. (3) We also compare CA-GAN with SMOTE, a naive but effective and popular resampling method, demonstrating superior performance of generative models in generalisation and synthesis of authentic data. (4) We show that CA-GAN is able to synthesise realistic data that can augment the real data, when used in a downstream predictive task.

## 2 Methods

### 2.1 Problem Formulation

Let $A$ be a vector space of features and let $a \in A$. Let $l$ be a binary mask, extracted from $L = \{0, 1\}$, a distribution modifier. Consider the following data set $D_0 = \{a_n\}_{n=1}^{N}$ with $l = 0$, with individual samples indexed by $n \in \{1, ..., N\}$ and $D_1 = \{a_m\}_{m=N+1}^{N+M}$ with $l = 1$, with individual samples indexed by $m \in \{N + 1, ..., N + M\}$ where $N > M$. Then, consider the dataset $D = D_0 \cup D_1$ as our training dataset. Notations inspired by [21].

**Our goals**. We want to learn a density $\hat{d}\{A\}$ that best approximates $d\{A\}$, the true distribution of $D$. We define $\hat{d}_1\{A\}$ as $\hat{d}\{A\}$ with $l = 1$ applied. From the modified distribution $\hat{d}_1\{A\}$ we draw random variables $X$ and add these to $D_1$ until $N = M$.

### 2.2 CGAN vs GAN

The Generative Adversarial Network (GAN) [22] entails 2 components, a generator and a discriminator. The generator $G$ is fed a noise vector $z$ taken from a latent distribution $p_z$ and outputs a sample of synthetic data. The discriminator $D$ inputs either fake samples created by the generator or real samples $x$ taken from the true data distribution $p_{data}$. Hence, the GAN can be represented by the following minimax loss function:

$$\min_{G} \max_{D} V(D, G) = \mathbb{E}_{x \sim p_{\text{data}}(x)}[\log D(x)] + \mathbb{E}_{z \sim p_z(z)}[1 - \log D(G(z))]$$

The goal of the discriminator is to maximise the probability to discern fake from real data, whilst the goal of the generator is to make samples realistic enough to fool the discriminator, i.e. to minimise $\mathbb{E}_{z \sim p_z(z)}[1 - \log D(G(z))]$. As a result of the reciprocal competition both the generator and discriminator improve during training.

The limitations of vanilla GAN models become evident when working with highly imbalanced datasets, where there might not be sufficient samples to train the models in order to generate minority class samples. A modified version of GAN, the Conditional GAN [23], solves this problem by using labels $y$, both in the generator and discriminator. The additional information $y$ divides the generation and the discrimination in different classes. Hence, the model can now be trained on the whole dataset, to then generate only minority class samples. Hence, the loss function is modified as follows:

$$\min_{G} \max_{D} V(D, G) = \mathbb{E}_{x \sim p_{\text{data}}(x)}[\log D(x|y)] + \mathbb{E}_{z \sim p_z(z)}[1 - \log D(G(z|y))]$$

GAN and CGAN, overall, share the same major weaknesses during training, namely mode collapse and vanishing gradient [24]. In addition, as GAN were initially designed to generate images, thus, they have been shown unsuitable to generate time-series [21] and discrete data samples [25].

## 2.3 CA-GAN vs WGAN-GP*

The WGAN-GP* introduced by Kuo et al. [18] solved many of the limitations posed by vanilla GANs. The model was a modified version of a WGAN-GP [16, 17], thus it applied the Earth Mover distance (EM) [26] to the distributions, which had been shown to solve both vanishing gradient and mode collapse [27]. In addition, the model applied Gradient Penalty during training, which helped to enforce more efficiently the Lipschitz constraint on the discriminator. More information on the WGAN-GP* architecture can be found in Appendix A.

We built our CA-GAN on the WGAN-GP* of Kuo et al. by conditioning the generator and the discriminator on static labels $y$. Hence, the updated loss functions used by our model are as follows:

$$L_D = \mathbb{E}_{z \sim p_z(z)}[D(G(z|y))] - \mathbb{E}_{x \sim p_{\text{data}}(x)}[D(x|y)] + \lambda_{GP}\mathbb{E}_{z \sim p_z(z)}[(||\nabla D(G(z|y))||_2 - 1)^2]$$

$$L_G = -\mathbb{E}_{z \sim p_z(z)}[D(G(z|y))] + \underbrace{\lambda_{corr}\sum_{i=1}^{n}\sum_{j=1}^{i-1}||r_{syn}^{(i,j)} - r_{real}^{(i,j)}||_{L_1}}_{\text{Alignment loss}}$$

Where $y$ can be any type of categorical label. During training the label $y$ were used to differentiate the minority from the majority class and during generation they were used to create fake samples of the minority class.

In comparison with WGAN-GP*, we also increased the number of biLSTMs from 1 to 3 both in the generator and the discriminator, as stacked biLSTMs have been shown to better capture complex time-series [28]. In addition we decreased learning rate and batch size during training. An overview of the CA-GAN architecture is shown in Figure 3.

## 3 Evaluation

Our dataset comprises 3343 hypotensive patients ([29]) admitted to critical care, the patients were either of Black (395) or Caucasian (2948) ethnicity. Each patient is represented by 48 data points, corresponding to the first 48 hours after the admission, in addition to 9 numeric, 4 categorical and 7 binary variables (20 in total) as shown in Table 3.

### 3.1 Evaluation Metrics

Evaluating the quality of the data produced by a generative model is anything but trivial. Several evaluation metrics have been proposed, but there is still no standardised evaluation method. In this work, given the complexity of the multivariate time series that we wanted to synthesise, we have chosen to adopt both a qualitative and quantitative evaluation of generated data. First, we used Maximum Mean Discrepancy (MMD) and Kullback–Leibler divergence to measure the difference between real and synthetic data for the underlying distribution of each variable. Second, we use Kendall rank correlation coefficient to evaluate the ability of the generative model to capture correlations between variables. Then, we verified that our model was generating authentic data (and not simply copying real data) by measuring the Euclidean Distance between real data and synthetic data. In this respect we also visualised real and synthetic data in a two dimensional latent space. Finally, we verified that our CA-GAN was able to generate useful new time series and able to capture the temporal correlation of the observations, by evaluating the predictive ability of an LSTM trained with synthetic data and evaluated on test data. Furthermore, several of our evaluation metrics were qualitatively analysed using plots of distributions, correlations, and two-dimensional representations of the datasets.

## 4 Results

In this section we present the comparison between the synthetic data generated by our CA-GAN and the data generated by WGAN-GP* and SMOTE (with 5-NN). We used each method to generate sufficient data to augment the minority class (Black patients) and balance the original dataset.

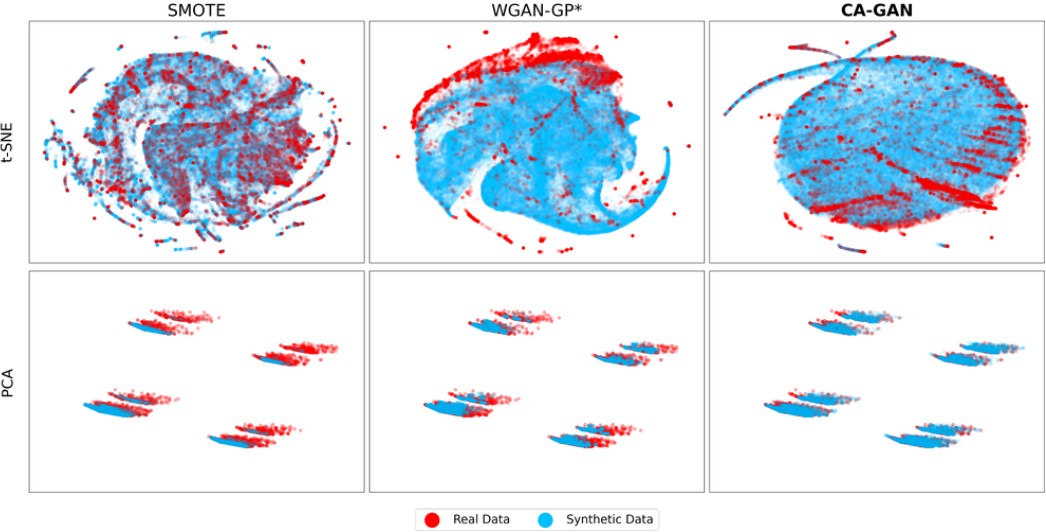

Figure 1: At the top, t-SNE two-dimensional representation of real and synthetic data for the three methods: SMOTE, WGAN-GP* and CA-GAN. It can be seen that CA-GAN provides a better coverage of the distribution of real data. At the bottom, PCA two-dimensional representation. CA-GAN provides the best coverage of the distribution of real data, followed by WGAN-GP* and SMOTE.

## 4.1 Qualitative evaluation

Before quantitatively analyzing the results of the three methods we are comparing, it is appropriate to show the data that has been generated using a visual approach, to provide an initial insight into the results obtained. t-SNE [30] allows us to plot real and synthetic datasets in a two dimensional latent space preserving the neighbourhood of data points, thus the real data appears differently in each plot, whereas this is not the case for UMAP in Figure 8.

Figure 1 shows the results of this representation. The data generated by WGAN-GP* remains almost entirely separate from the real data, indicating that this method is not able to capture the underlying structure of the real data. On the other hand, CA-GAN and SMOTE generate synthetic data that overlaps with real data. However, since SMOTE data points are an interpolation of real data, they create a pattern in which they fill the spaces between the closest points, without expanding into the embedded space. Instead, CA-GAN data points are spread homogeneously in space, while remaining within the confines of the real distribution. This is an indication of the ability of our model to better generalise in the data space, resulting in authentic data.

Subsequently, the use of PCA that attempts to preserve the global structure (in contrast to t-SNE), shows that CA-GAN is able to generate data points that cover the entire variance of the real data, while SMOTE and WGAN-GP* tend to converge on the mean, flattening their variance as shown in Figure 1. Finally, in the appendix we also present UMAP latent representation of the data (Figure 8) and we show in more detail the distributions of each variable generated with the three methods superimposed on the real data (Figures 4, 5, 6).

## 4.2 Quantitative evaluation

For each variable $v$ of the dataset, the Kullback-Leibler (KL) divergence[31] measures the similarity between the discrete density function of the real data and that of the synthetic data: $D_{KL}(P_v \| Q_v) = \sum_i P_v(i) \log \frac{P_v(i)}{Q_v(i)}$. The smaller the divergence, the more similar the distributions, with zero for identical distributions. Table 1 shows the results of the KL divergence for each variable for a single run. Our model has the lowest median across all variables. CA-GAN has better results than WGAN-GP* and SMOTE overall. It should be noted that the latter is an algorithm designed specifically to maintain the distribution of the original variables.

Table 1: KL-Divergence and Maximun Mean Discrepancy between the distribution of real and synthetic data for each variable.

| | KL-divergence | | | MMD | | |
|---|---|---|---|---|---|---|
| | SMOTE | WGAN-GP* | **CA-GAN** | SMOTE | WGAN-GP* | **CA-GAN** |
| MAP | 0.11182 | 0.24941 | 0.17164 | 0.00137 | 0.00824 | 0.00110 |
| Diastolic BP | 0.28191 | 0.91622 | 0.24342 | 0.00155 | 0.00209 | 0.00086 |
| Systolic BP | 0.06405 | 0.10588 | 0.13194 | 0.00138 | 0.00120 | 0.00092 |
| Fluid Boluses | 0.01121 | 0.00358 | 0.00052 | 0.00047 | 0.00022 | 0.00003 |
| Urine | 0.00892 | 0.15183 | 0.00901 | 0.01321 | 0.08567 | 0.08443 |
| Vasopressors | 0.03622 | 0.05955 | 0.00175 | 0.00463 | 0.00883 | 0.00031 |
| ALT | 0.00068 | 0.37020 | 0.00800 | 0.01356 | 0.20156 | 0.18616 |
| AST | 0.00083 | 0.18162 | 0.00455 | 0.01323 | 0.20920 | 0.19538 |
| FiO2 | 0.00858 | 0.01950 | 1.36841 | 0.00091 | 0.00043 | 0.00012 |
| GCS | 0.02432 | 0.02571 | 0.01934 | 0.05206 | 0.00688 | 0.00791 |
| PO2 | 0.00315 | 0.13503 | 0.31726 | 0.00992 | 0.25091 | 0.24806 |
| Lactic Acid | 0.03192 | 0.42781 | 0.45402 | 0.01084 | 0.16273 | 0.19777 |
| Serum Creatinine | 0.02079 | 0.02851 | 0.08827 | 0.01892 | 0.22812 | 0.03313 |
| Urine (M) | 0.19717 | 0.00279 | 0.00070 | 0.09954 | 0.00170 | 0.00043 |
| ALT/AST (M) | 0.01872 | 0.00027 | 0.00031 | 0.00050 | 0.00001 | 0.00002 |
| FiO2 (M) | 0.07361 | 0.00965 | 0.00459 | 0.00892 | 0.00224 | 0.00103 |
| GCS_total (M) | 0.12043 | 0.00072 | 0.00013 | 0.03776 | 0.00030 | 0.00006 |
| PO2 (M) | 0.03846 | 0.00751 | 0.00033 | 0.00238 | 0.00067 | 0.00003 |
| Lactic Acid (M) | 0.03962 | 0.00010 | 0.00136 | 0.00274 | 0.00001 | 0.00015 |
| Serum Creatinine (M) | 0.05844 | 0.00777 | 0.00005 | 0.00613 | 0.00117 | 0.00001 |
| Median | 0.03407 | 0.02711 | **0.00629** | 0.00752 | 0.00217 | **0.00089** |

Using Maximum Mean Discrepancy (MMD)[32], we calculated the distance between the distributions based on kernel embeddings of distributions, that is, the distance of the distributions represented as elements of a reproducing kernel Hilbert space (RKHS). We used a Radial Basis Function (RBF) Kernel: $K(x_{real}, x_{syn}) = \exp\left(-\frac{\|x_{real} - x_{syn}\|^2}{2\sigma^2}\right)$, with $\sigma = 1$. The right half of Table 1 shows the MMD results for SMOTE, WGAN-GP* and our CA-GAN, where the latter shows the best median performance across all the variables.

## 4.3 Correlations

We used the Kendall rank correlation coefficient $\tau$ [33] to investigate whether synthetic data maintained original correlations between variables found in the real data. This choice is motivated by the fact that $\tau$ coefficient does not assume a normal distribution, that some of our variables do not have, as shown in Figures 4, 5, 6. Figure 7 shows the results of Kendall's rank correlation coefficients. Comparing them with real data, CA-GAN (Figure 2) captures the original correlations, as does SMOTE, with the former having closest results on categorical variables, and the latter on numerical ones. Once again WGAN-GP* shows the lowest performance, accentuating correlations that do not exist in real data.

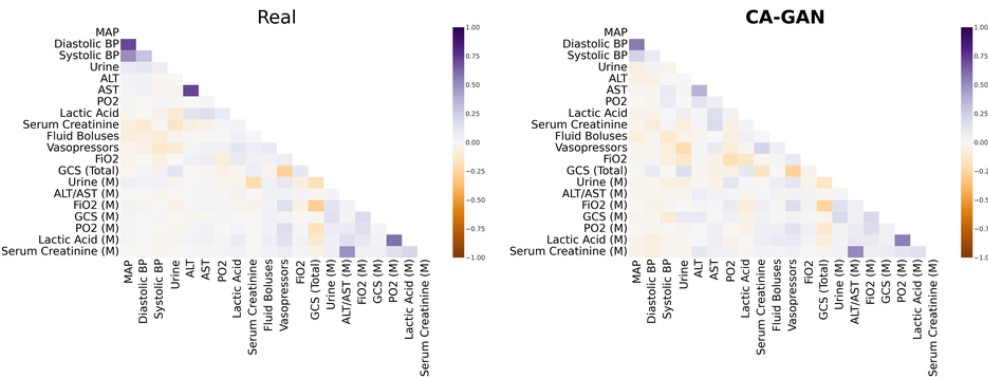

Figure 2: Kendall's rank correlation coefficients for the real data and the data generated with CA-GAN.

### 4.4 Authenticity

When generating synthetic data, it is important that the output is a realistic representation of the original data, but also need to verify that the model has not learned to copy the real data. While unlikely, GANs can overfit by memorizing real data [34]. In order to evaluate the originality of the output of our model we use Euclidean Distance ($L_2$ Norm). The shortest distance between a synthetic sample and a real one is $3.14$. This result, coupled with the visual representation of CA-GAN (shown in Figure 1), shows the ability of our model to produce authentic data. SMOTE, on the other hand, which by design interpolates the original data points, is unable to explore the underlying multidimensional space, as the minimum Euclidean distance of its generated data samples is $0.00234$. Since the goal of our work is the augmentation of an existing dataset, we did not consider aspects related to privacy preservation and adversarial attacks.

### 4.5 Downstream regression task

Finally, we wanted to evaluate the ability of CA-GAN to maintain the temporal properties of time series data. Furthermore, since we have set ourselves the objective of augmenting the minority class to mitigate data poverty, we want to verify that the new augmented dataset, generated with our model, is able to maintain or improve the predictive performance on a downstream task. Hence, we trained a Bidirectional LSTM, first only with real data as the baseline, and later with the synthetic dataset and also the augmented dataset. The LSTM takes 20 hours of data as an input and provides a prediction on the subsequent 1 hour, in a sliding window fashion. To ensure the fairness of our result, the time series data points of 60 black patients (representing 15% of the overall data) were put apart as a test set and were used to evaluate the performance in a regression task.

Table 2 shows the mean relative errors between the LSTM prediction and the actual observations, for the model trained only with real data, the one trained with only synthetic data (CA-GAN) and a model trained with augmented data (both real and synthetic), which would be used in a downstream regression or classification task.

It should be noted that relative errors in fluid bolus, urine and vassopressors are particularly high in comparison to the other variables due to the challenge in prediction of these variables in general (stemming in part from the manner in which they are collected and recorded), rather than any issue inherent to the synthetic data. The prediction errors for these two variables are also consistent with SMOTE and WGAN-GP*.

Table 2: Mean prediction errors of a biLSTM trained on real, synthetic and augmented data.

|  | Real | Synthetic | **Augmented** |
|---|---|---|---|
| MAP | 13.35 | 11.57 | 11.19 |
| Diastolic BP | 18.21 | 13.27 | 14.99 |
| Systolic BP | 9.21 | 15.19 | 9.56 |
| Fluid Boluses | 61.93 | 70.41 | 75.99 |
| Urine | 28.41 | 26.92 | 28.22 |
| Vasopressors | 37.87 | 40.91 | 36.43 |
| ALT | 4.22 | 10.46 | 7.71 |
| AST | 15.07 | 11.27 | 7.38 |
| FiO2 | 2.59 | 2.64 | 6.91 |
| GCS | 3.07 | 2.60 | 4.08 |
| PO2 | 3.53 | 4.09 | 5.02 |
| Lactic Acid | 10.72 | 13.02 | 6.58 |
| Serum Creatinine | 11.92 | 10.00 | 3.65 |
| Median | 11.92 | 11.57 | **7.71** |

## 5 Conclusions and future work

In this work we have presented and evaluated Conditional Augmentation GAN (CA-GAN), an architecture that can overcome some of the shortcomings of the current approaches (WGAN-GP* and SMOTE) in augmenting the minority class of an imbalanced dataset. Through qualitative and quantitative evaluation we have shown that CA-GAN can generate authentic samples with greater distribution coverage than other approaches we evaluated, while ensuring that synthetic data are not merely copies of the real data with substantial distances between them. Furthermore, we have shown that augmenting the dataset with the synthetic data generated by CA-GAN can lead to lower relative errors in the prediction task, indicating that our model is able to generalise well from the original data. Furthermore, our approach can make use of the overall dataset, and not only the minority class as is the case with WGAN-GP* and SMOTE, thus being applicable also in presence of extremely imbalanced datasets, such as rare diseases. In the future we plan to evaluate the performance of our architecture with other datasets and also in the presence of other ethnicities with even lower data representation, as well as in cases where classes are represented by categorical variables, or continuous variables, becoming a regression problem for the latter.

# 6 Acknowledgements

The authors thank Nicholas I-Hsien Kuo for their helpful comments in improving the paper and Gabriele Franch for the UMAP suggestions.

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

# Appendix A  Analysis of WGAN-GP* architecture

In contrast with vanilla WGAN-GP, WGAN-GP* employed soft embeddings [35, 36], which allowed the model to use inputs as numeric vectors for both binary and categorical variables, and a Bidirectional LSTM layer [37, 38], which allowed for the generation of samples in time-series. With regard to the loss functions, while $L_D$ was kept the same, $L_G$ was modified by Kuo et al. [18] by introducing alignment loss, which helped the model to better capture correlation among variables over time. Hence, the loss functions of WGAN-GP* are the following:

$$L_D = \mathbb{E}_{z \sim p_z(z)}[D(G(z))] - \mathbb{E}_{x \sim p_{\text{data}}(x)}[D(x)] + \lambda_{GP}\mathbb{E}_{z \sim p_z(z)}[(\|\nabla D(G(z))\|_2 - 1)^2]$$

$$L_G = -\mathbb{E}_{z \sim p_z(z)}[D(G(z))] + \lambda_{corr}\underbrace{\sum_{i=1}^{n}\sum_{j=1}^{i-1}\|r_{syn}^{(i,j)} - r_{real}^{(i,j)}\|_{L_1}}_{\text{Alignment loss}}$$

To calculate alignment loss it was first computed Pearson's r correlation [39] for every unique pair of variables $X^{(i)}$ and $X^{(j)}$. The $L_1$ loss was then applied to the differences in the correlations between $r_{syn}$ and $r_{real}$, with $\lambda_{corr}$ representing a constant acting as a strength regulator of the loss.

# Appendix B  Proposed architecture of our CA-GAN

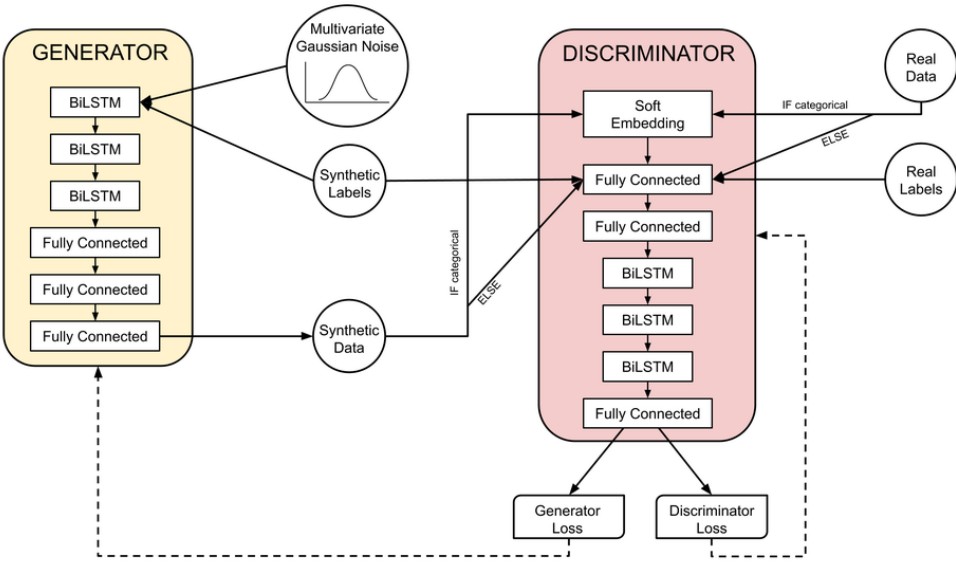

Figure 3: Proposed architecture of our CA-GAN.

# Appendix C    Comparison of distributions and correlations

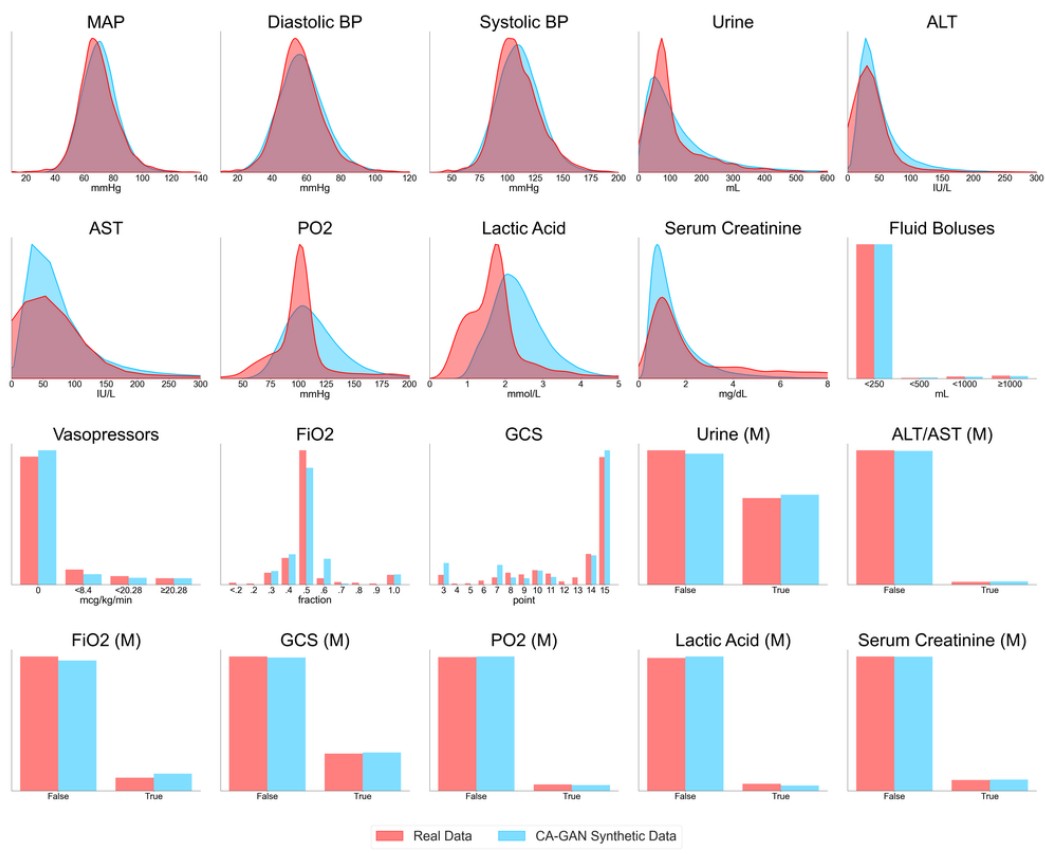

Figure 4: Overlaid distribution plots of real data and CA-GAN synthetic data.

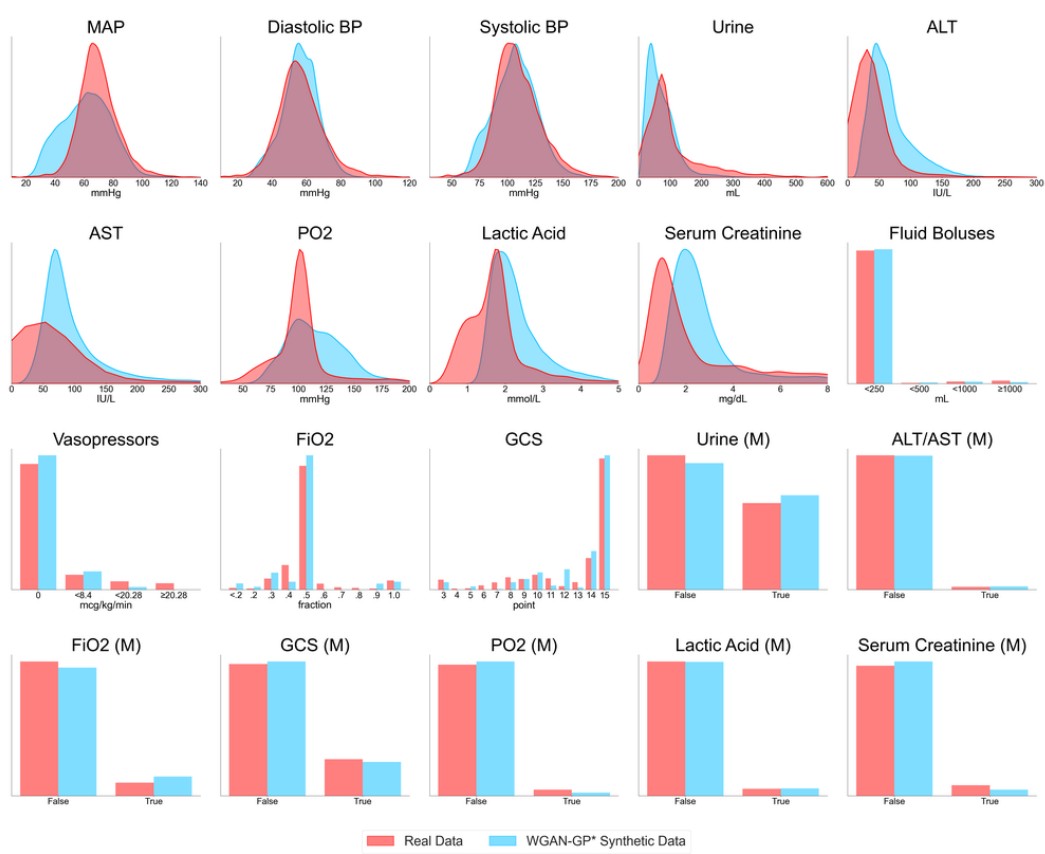

Figure 5: Overlaid distribution plots of real data and WGAN-GP* synthetic data.

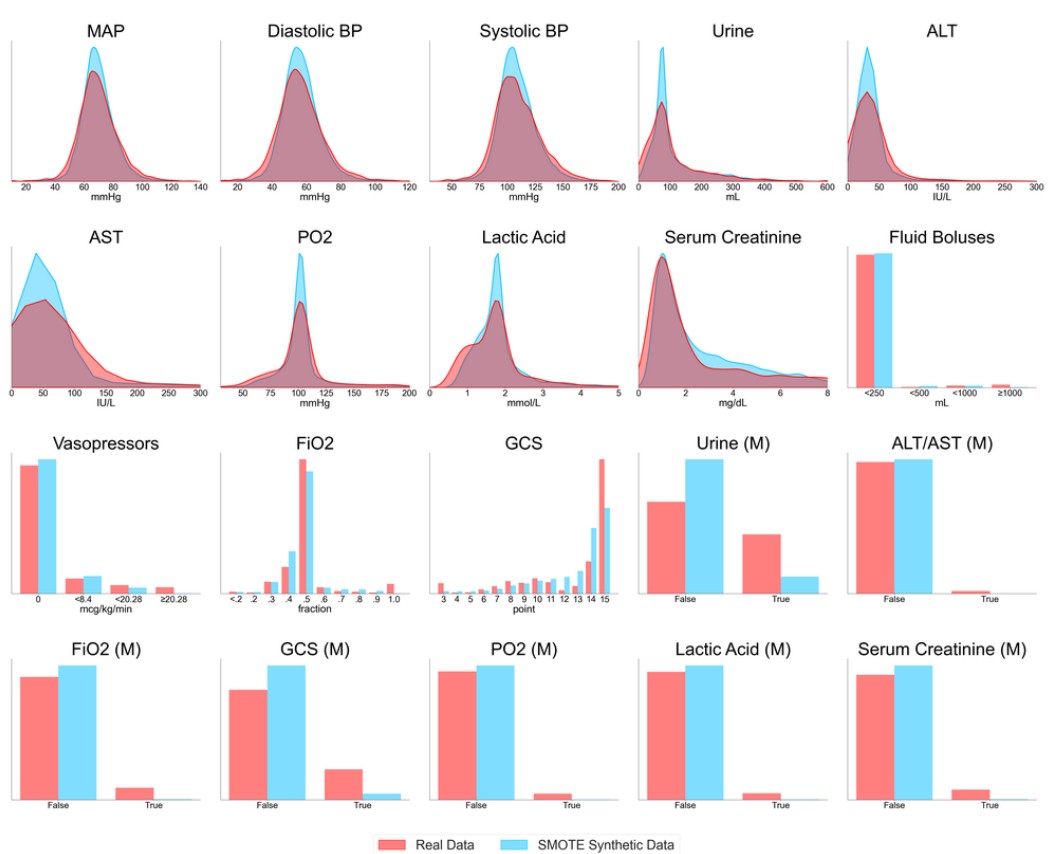

Figure 6: Overlaid distribution plots of real data and SMOTE synthetic data.

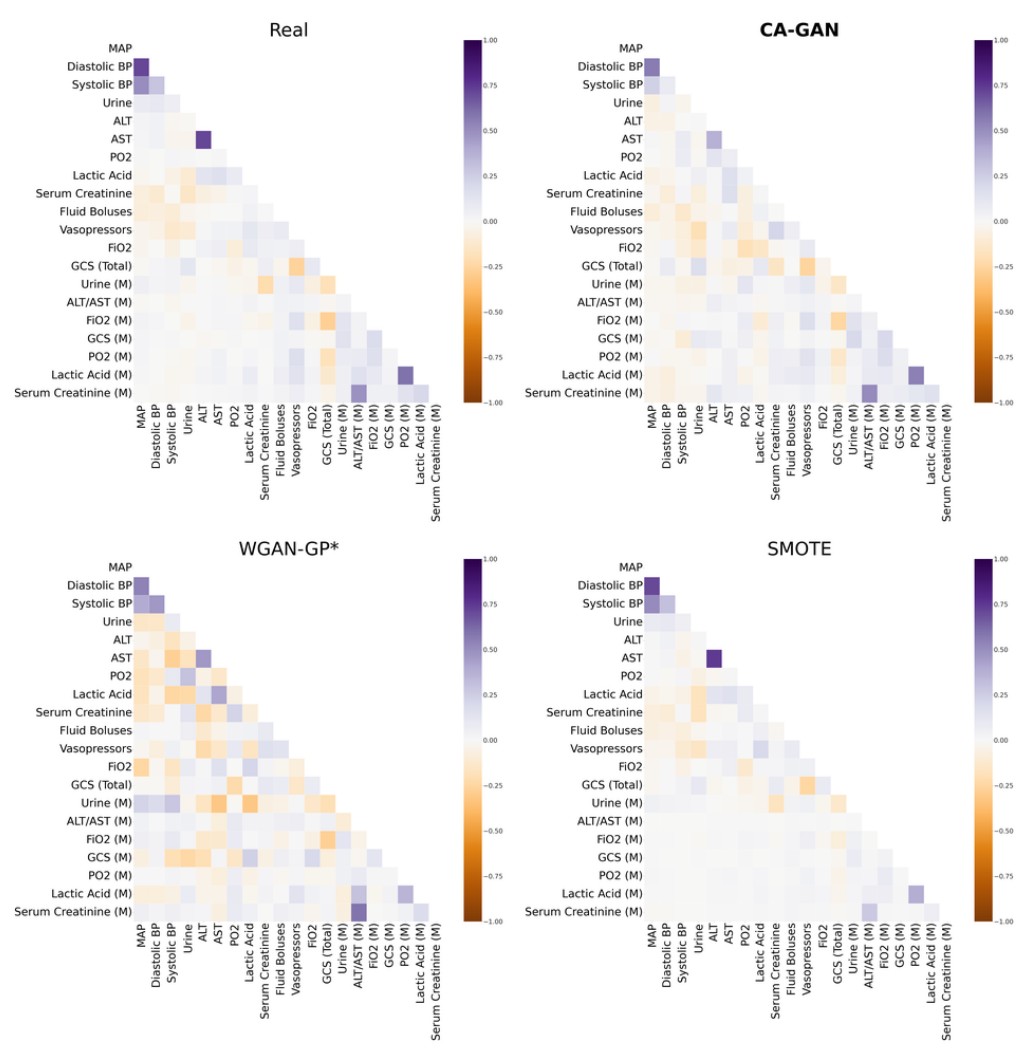

Figure 7: Kendall's rank correlation coefficients for real data and synthetic data generated with CA-GAN, WGAN-GP* and SMOTE.

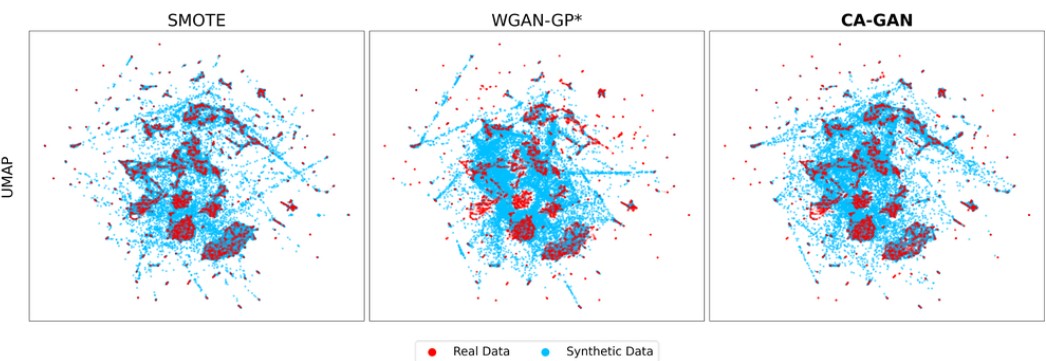

Figure 8: UMAP two-dimensional representation of real and synthetic data, comparing the three methods: SMOTE, WGAN-GP* and CA-GAN.

# Appendix D   Dataset

Table 3: Variables of the hypotension dataset [18], used in our evaluation.

| Variable Name | Data Type | Unit | Descriptive Statistics | |
|---|---|---|---|---|
| Mean Arterial Pressure (MAP) | numeric | mmHg | Median: 65.34 (Q1: 59.30, Q3: 71.19) | |
| Diastolic Blood Pressure (BP) | numeric | mmHg | Median: 54.33 (Q1: 48.37, Q3: 60.26) | |
| Systolic BP | numeric | mmHg | Median: 113.21 (Q1: 104.23, Q3: 121.60) | |
| Urine | numeric | mL | Median: 106.21 (Q1: 68.92, Q3: 164.23) | |
| Alanine Aminotransferase (ALT) | numeric | IU/L | Median: 32.55 (Q1: 24.59, Q3: 46.09) | |
| Aspartate Aminotransferase (AST) | numeric | IU/L | Median: 46.82 (Q1: 35.81, Q3: 67.75) | |
| Partial Pressure of Oxygen (PaO2) | numeric | mmHg | Median: 103.02 (Q1: 91.34, Q3: 114.66) | |
| Lactate | numeric | mmol/L | Median: 1.50 (Q1: 1.29, Q3: 1.80) | |
| Serum Creatinine | numeric | mg/dL | Median: 1.11 (Q1: 0.83, Q3: 1.62) | |
| Fluid Boluses | categorical | mL | 4 Classes | |
| | | | [0,250) : 97.32%; | [250,500) : 0.28% |
| | | | [500,1000) : 1.46%; | $\geq$ 1000 : 0.94% |
| Vasopressors | categorical | mcg/kg/min | 4 Classes | |
| | | | 0 : 84.14%; | (0,8.4) : 8.34% |
| | | | [8.4,20.28) : 3.68%; | $\geq$ 20.28 : 3.83% |
| Fraction of Inspired Oxygen (FiO2) | categorical | fraction | 10 Classes | |
| | | | $\leq$ 0.2 : 0.00%; | 0.2 : 0.54% |
| | | | 0.3 : 2.84%; | 0.4 : 10.85% |
| | | | 0.5 : 63.30%; | 0.6 : 8.58% |
| | | | 0.7 : 1.32%; | 0.8 : 0.20% |
| | | | 0.9 : 2.63%; | 1.0 : 9.75% |
| Glasgow Coma Scale Score (GCS) | categorical | point | 13 Classes | |
| | | | 3 : 6.61% | 4 : 2.16% |
| | | | 5 : 0.00% | 6 : 3.00% |
| | | | 7 : 4.77% | 8 : 0.00% |
| | | | 9 : 2.22% | 10 : 4.32% |
| | | | 11 : 2.46% | 12 : 3.56% |
| | | | 13 : 1.00% | 14 : 9.80% |
| | | | 15 : 60.09% | |
| Urine Data Measured (Urine (M)) | binary | - | False: 63.07% | True: 36.93% |
| ALT or AST Data Measured (ALT/AST (M)) | binary | - | False: 98.50% | True: 1.50% |
| FiO2 (M) | binary | - | False: 92.49% | True: 7.51% |
| GCS (M) | binary | - | False: 81.49% | True: 18.51% |
| PaO2 (M) | binary | - | False: 97.56% | True: 2.44% |
| Lactic Acid (M) | binary | - | False: 96.98% | True: 3.02% |
| Serum Creatinine (M) | binary | - | False: 95.26% | True: 4.74% |

