# OpenReview forum: "Mitigating Health Data Poverty: Generative Approaches versus Resampling for Time-series Clinical Data"
_NeurIPS.cc/2022/Workshop/SyntheticData4ML — Neurips 2022 SyntheticData4ML_

### Official Review · Reviewer_8jon · 2022-10-11
**The paper presents a methodology for addressing, class imbalance and data augmentation, very relevant problem in healthcare datasets. The work is easy to follow, yet the evaluation could be improved with slight changes.**

**Rating:** 9
**Confidence:** 4

**Review:**

Pros:
1) The authors present a simple yet valuable modification to WGAN-GP*, specifically trained for data augmentation purposes.
2) The work is clearly explained and easy to follow and compares with SMOTE, a popular method used in clinical and epidemiological studies using datasets similar to MIMIC.
3) The authors show both qualitative and quantitive evaluation, which is often compromised in works in this area of research.


Cons:
1) While the authors evaluate the overall distance between the synthetic and real datasets, this might not demonstrate the model’s true robustness against adversarial membership and /or attribute inference attacks.  I recommend the authors consider expanding their privacy preservation evaluation.

---

### Official Review · Reviewer_fMVD · 2022-10-18
**A well-motivated paper with promising results but slightly suffers from lack of consistency and error bars**

**Rating:** 6
**Confidence:** 3

**Review:**

Summary: The work consists of a new type of GAN, CA-GAN, that aims to improve generation of minority-class samples from the training set. The authors evaluate the method on a hospital dataset with common comparison methods that are prone to suffer from algorithmic bias.

The literature review is good with a concise overview of related work. The problem is well structured and methods well described. The visualisations are good, but confusing as they are inconsistent between methods. The authors also test their method on a downstream regression task.

Pros
- Clear motivation to reduce bias and improve accuracy in real-world data problems
- Well written with comprehensive evaluation across different metrics and downstream tasks

Cons and questions
- Visualisations need clarifications. In Figure 1, why are the embedded real data visualisations different for the three models? Surely the real data is independent of the model used to generate the synthetic data.
- Are the results from a single run or average of multiple trained models? Without errors on the means it’s hard to judge the significance of the difference between methods
- The experiments show the benefit of CA-GAN over other methods for data accuracy on the downstream task of the minority class, what was the overall performance? Did it also outperform?
- In 2.1: abe \in A, should that be just a?

---

### Meta-Review · Area_Chair_JZst · 2022-10-19

**Recommendation:** Accept